# TST[R]: Target Similarity Tuning Meets the Real World

**Anirudh Khatry**[*]
**Priyanshu Gupta**
**Ananya Singha**
Microsoft
Bangalore, India

**Sumit Gulwani**
**Vu Le**
Microsoft
Redmond, US

**Mukul Singh**
Microsoft
Delhi, India

**Gust Verbruggen**
Microsoft
Keerbergen, Belgium

## Abstract

Target similarity tuning (TST) is a method of selecting relevant examples in natural language (NL) to code generation through large language models (LLMs) to improve performance. Its goal is to adapt a sentence embedding model to have the similarity between two NL inputs match the similarity between their associated code outputs. In this paper, we propose different methods to apply and improve TST in the real world. First, we replace the sentence transformer with embeddings from a larger model, which reduces sensitivity to the language distribution and thus provides more flexibility in synthetic generation of examples, and we train a tiny model that transforms these embeddings to a space where embedding similarity matches code similarity, which allows the model to remain a black box and only requires a few matrix multiplications at inference time. Second, we how to efficiently select a smaller number of training examples to train the TST model. Third, we introduce a ranking-based evaluation for TST that does not require end-to-end code generation experiments, which can be expensive to perform.

## 1 Introduction

Code generation from natural language utterances is an important and useful ability of large language models (LLMs). Experienced developers can save time, and less experienced users can use natural language to perform data transformation tasks that they would otherwise have to carry out manually (Liu et al., 2023). Improving the code generation capabilities of LLMs is a popular research area (Wang et al., 2021).

Target similarity tuning (TST) was proposed as a method for selecting relevant examples to exploit the in-context learning ability of LLMs and improve performance (Poesia et al., 2022).

---
[*] First author

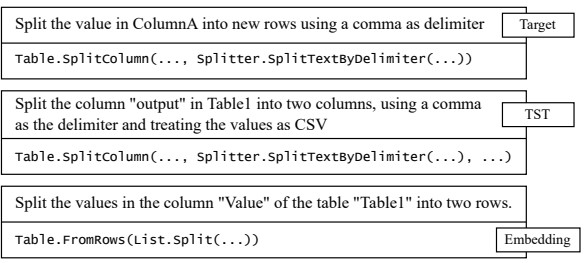

Figure 1: Example of a target utterance and associated code, the examples selected by TST and by relying on default embeddings.

TST improves the alignment between sentence embeddings of utterances by fine-tuning their cosine similarities to match the similarity of their associated code snippets. Sentence-BERT (Reimers and Gurevych, 2019) is used as embedding model. Especially for rare programming languages, the model benefits from seeing relevant code (+3% for SQL versus +16% for SMCalFlow using GPT-3).

**Example.** *Figure 1 shows an example of how TST improves over default embeddings by teaching the model which parts of the utterance are important. The embeddings focus heavily on the "into rows" part of the utterance (where the user might have made a mistake). With TST, we can teach it to focus on the "delimiter" part by focusing on the similarity between code snippets.*

This paper addresses four limitations of TST when applying it in the real world: (1) sensitivity to language, (2) inference with transformer models, (3) dataset curation and (4) evaluation.

Limitations (1) and (2) are due to the sentence embedding model. Training utterances and real utterances often come from different distributions, for example, from users with different skill levels, and smaller models might be unable to capture this variation. When hosted, third-party LLMs are used for code generation, performing inference with transformer models might not be possible, making TST

hard to use in production.

We address both these challenges by replacing the sentence transformer with embeddings from a hosted, third-party model and training a fully connected neural network (FCNN) to transform the embeddings to capture the code similarity. The large model provides stronger embeddings and the FCNN only requires a (few) matrix-vector multiplication during inference.

Dataset curation (3) is important, as given $n$ (utterance, code) pairs, we can sample $n(n-1)/2$ pairs to create training examples for TST, most of consist of irrelevant code pairs. Since we care about distinguishing the best examples, these irrelevant pairs are not desired. We address this challenge by selecting positive and negative examples close to the important decision boundary and create a smaller dataset that yields better performance and is much faster to train on.

Our method proposed method of training the FCNN on top of frozen embeddings on a relevant set of examples is called $\text{TST}^\text{R}$ (*tastier*), where R stands for real world.

Finally, optimizing hyperparameters of the TST model is expensive to evaluate when LLM calls are required. We show that an evaluation based on ranking of examples close to the decision boundary matches the end-to-end performance, providing a cheap way to evaluate TST models.

We make the following contributions:

(1 + 2) We show that training a small model on top of frozen embeddings makes $\text{TST}^\text{R}$ easier to train and use, and less sensitive to variations in language.

(3) We show that selecting examples close to the important decision boundary allows us to train a TST model with much fewer examples.

(4) We show that ranking (train, test) utterance pairs correlates to the performance of end-to-end code generation, providing a cheap way to evaluate TST models.

## 2   Related Work

Code generation from natural language is a popular area of research (Le et al., 2022; Li et al., 2022). Instead of starting from scratch, fine-tuning a pre-trained natural language understanding model to generate code is a popular approach, for example,

CodeBERT (Feng et al., 2020) was trained from BERT, CodeT5 (Wang et al., 2021) was trained from T5 and Codex (Chen et al., 2021) was trained from GPT-3.

A powerful method to generate code from natural language is prompting large language models (Chen et al., 2021). As opposed to fine-tuning, this does not require large training datasets and expensive compute. Few-shot prompting consistently improves the performance of LLMs across a variety of tasks (Brown et al., 2020). Besides helping the model pick the correct programming language (Athiwaratkun et al., 2023) the provided few-shots can teach the model about specific functions or parameters and contextualization.

One way of selecting relevant examples is by using sentence embeddings (Liu et al., 2021). In some cases, however, similar natural language does not correspond to similar code, and vice versa. Synchromesh (Poesia et al., 2022) introduced target similarity tuning (TST) to address this challenge and fine-tunes the sentence embedding similarity to match the associated code similarity.

This work builds on the concept of TST and improves on important implementation details for training (selecting examples and allowing synthetic data generation), evaluating (cheap evaluation with a proxy metric) and deploying (API call and small FCNN do not have hardware requirements) TST in practice.

## 3   $\text{TST}^\text{R}$

We briefly recap TST, how that transfers to $\text{TST}^\text{R}$ and how examples closer to the relevant decision boundary are selected to improve training

### 3.1   TST

Given two (utterance, code) pairs $(u_1, c_1)$ and $(u_2, c_2)$, vanilla TST fine-tunes a semantic textual similarity (STS) model $S_m$ to minimize

$$\|S_m(u_1, u_2) - S_c(c_1, c_2)\| \tag{1}$$

with $S_c$ a similarity between code pairs. The STS model is SBERT, which pools BERT tokens and fine-tunes pooled embeddings to capture similarity between pairs of sentences. TST then transforms the embedding to capture properties of utterances that make their associated code similar.

## 3.2 TST with Embeddings

We decouple the embedding from the similarity and compute

$$S_m(u_1, u_2) = \cos(t_\theta(m(u_1)), t_\theta(m(u_2))) \quad (2)$$

with $\cos$ the cosine similarity between vectors, $m$ a $d$-dimensional embedding model (such as SBERT) and $t_\theta : \mathbb{R}^d \rightarrow \mathbb{R}^{d'}$ a trainable transformation. The parameters of $m$ are frozen. To keep $t_\theta$ simple, we use a fully connected network with $\tanh$ activation.

## 3.3 Training

The training data consists of $(u_1, u_2, S_c(c_1, c_2))$ triplets. Instead of randomly sampling $(u_1, c_1)$ and $(u_2, c_2)$ pairs, we aim to find examples close to the relevant decision boundary. That is, we care about examples for which: (1) $S_c(c_1, c_2)$ is high, or (2) $S_c(c_1, c_2)$ is low but $\cos(m(u_1), m(u_2))$ is high—that are similar in the original embedding space but have dissimilar code. In other words, we care about examples with properties that we need to learn and that we need to unlearn. For each pair $(u_i, c_i)$ we therefore rank all other $(u_j, c_j)$ by $S_c(c_i, c_j)$ and select the top-$\lambda_k$ best ones. We then skip $\lambda_s$ examples to ensure that code similarity is not too high, rank the remaining examples by $S_c(m(u_i), m(u_j))$ and again select the top-$\lambda_k$ best ones in this new ranking. $\lambda_k$ and $\lambda_s$ are hyperparameters.

## 4 Evaluation Setup

This section describes the datasets, metrics, models and hyperparameters used in our experiments.[1]

### 4.1 Datasets

We evaluate $\text{TST}^R$ on three NL-to-code datasets across different low-resource languages. For each language, we also report the evaluation metric and code similarity $S_c$.

#### 4.1.1 Power Query M

The Power Query M language (or M) is used for transforming data in Power Query. We use the data from Khatry et al. (2023) consisting of code snippets sourced from StackOverflow (test) and the Power Query Community Forum[2] (train and test). The testing set contains 500 snippets annotated

by experts. The training set includes 8000 snippets annotated via an LLM (text-davinci-002). We report *execution match* and *sketch match*, two standard metrics for code generation (Poesia et al., 2022; Singh et al., 2022). Execution match (boolean) is determined by executing both the ground truth and generated code snippets on a table and checking for equality. Sketch match ($\in [0, 1]$) is computed as the normalized (Levenshtein) edit similarity between the ground truth and generated code snippets after masking constants (strings and numbers) and identifiers (column names). We use sketch match as $S_c$.

#### 4.1.2 SMCalFlow

SMCalFlow is a task-oriented dialogue dataset of user-agent conversations, where each user query is annotated with a program in a domain-specific language that facilitates a dialogue over a dataflow graph (Andreas et al., 2020). In line with previous work (Poesia et al., 2022), we select the first turn from each dialogue to define an NL-to-code task and sample 2000 training examples (out of 40K). The test set consists of 2673 examples. We follow Poesia et al. (2022) and use the normalized edit similarity for both evaluation and $S_c$.

#### 4.1.3 Bash

The nl2bash dataset consists of bash code snippets, each with an expert-curated natural language description (Lin et al., 2018). The train and test sets contain 8090 and 606 examples, respectively. We use the *template match* metric proposed in the original paper for both evaluation and $S_c$.

### 4.2 Models and Hyperparameters

We use text-embedding-ada-002 (ada) and text-davinci-003 (GPT-3) (both from OpenAI) as the embedding and code generation models. In $\text{TST}^R$, we use two fully connected layers with 512 parameters (see Section 5.2). To prevent overfitting, we apply dropout (0.3) between the embedding and the fully connected layer. $\lambda_k$ and $\lambda_s$ are set to 4.

### 4.3 Baselines

Across experiments, we use the following baselines for example selections with embeddings. For each baseline, we select eight examples.

- Vanilla ada and SentenceBERT embeddings.

- TST (Poesia et al., 2022) trained on examples selected according to our selection strategy

---

[1]https://github.com/microsoft/prose-benchmarks/tree/main/TSTR

[2]https://community.fabric.microsoft.com/t5/Power-Query/bd-p/power-bi-services

Table 1: TST$^R$ performance across languages on end-to-end code generation task. We find that TST easily overfits on the language distribution of the training data, but TST$^R$ does not.

| Technique | M | SMCF | Bash |
|---|---|---|---|
| Static | 0.20 | 0.43 | 0.56 |
| SentenceBERT | 0.53 | 0.87 | 0.65 |
| ada | 0.54 | 0.89 | 0.67 |
| TST | 0.52 | 0.89 | 0.63 |
| TST$^R$ | **0.55** | **0.90** | **0.68** |

(Section 3.3) for one epoch. Using random examples performed worse.

- A hybrid approach with frozen SentenceBERT embeddings instead of ada (called TST$^f$).

## 5 Evaluation

We perform experiments to compare TST$^R$ against baseline embedding retrieval methods (5.1), we show that ranking relevant examples serves as a proxy metric to optimize hyperparameters without LLMs (5.2), we evaluate how embeddings from large models (ada) are more robust with respect to variations in language (5.3), and we show the effect of selecting relevant examples to train TST$^R$ (5.4).

### 5.1 Performance

Table 1 shows results of TST$^R$ and baselines, as well as a static prompt with eight randomly selected examples. TST$^R$ consistently performs better (+1%) over vanilla embeddings.

Surprisingly, the original TST approach to fine tune SentenceBERT hurts performance on M (-1%) and Bash (-2%). This may be attributed to overfitting on the language of the training set, and not being able to relate new variations in language to the code similarity (see Section 5.3).

### 5.2 Standalone Evaluation

We create a pairwise ranking dataset to evaluate TST$^R$ without performing end-to-end code generation and evaluate this approach on M.

Each test point consists of a triplet $(u_r, u_p, u_n)$ where $u_r$ is a reference utterance from the testing dataset, $u_p$ and $u_n$ are candidate utterances from the training dataset, and $S_c(c_p, c_r) > S_c(c_n, c_r)$. We consider two ways of sampling $u_p$ and $u_n$ and thus create two testing datasets: at random ($\star$) or close to the relevant decision boundary, similar to

how the training dataset is created ($\bullet$). We count proportion of correct pairwise decisions.

We compare the execution match (end-to-end) and pairwise ranking evaluation for different embedding-based example selection strategies in Figure 2. Besides baselines, we also consider the theoretical maximum for a given similarity using the code–code similarity.

Our ranking evaluation captures the alignment of the TST model with relevant examples, the ones that are the most similar to a target code snippet, observed by the distinct relation between the trained TST models ($\bullet$) and the theoretical maximum. This relation does not hold when considering randomly sampled negative examples ($\star$). Embeddings rank poorly for relevant examples, but very good for random examples. These observations highlight the need to select relevant examples for standalone ranking: some nuances of similarities in natural language should be *unlearned*.

Figure 3 shows end-to-end and ranking results for different configurations of TST$^R$ on the $\bullet$ benchmark. A model with too few parameters does not learn enough, and too large models (likely) overfit. More interesting is the relation between ranking performance and execution match, where we can use the former as a proxy to determine the number of parameters of TST$^R$.

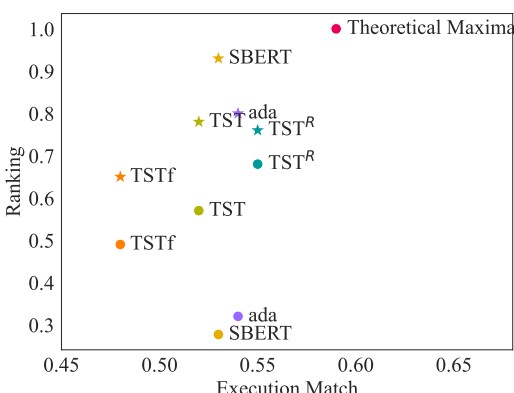

Figure 2: Relation between execution match (top-1) and our pairwise ranking evaluation with random ($\star$) and relevant ($\bullet$) negative examples on the M dataset. Ranking with relevant examples shows a relation with code generation performance.

### 5.3 Variation in Language

We show how TST$^R$ handles variations in language by creating three different testing datasets, with $u_r$

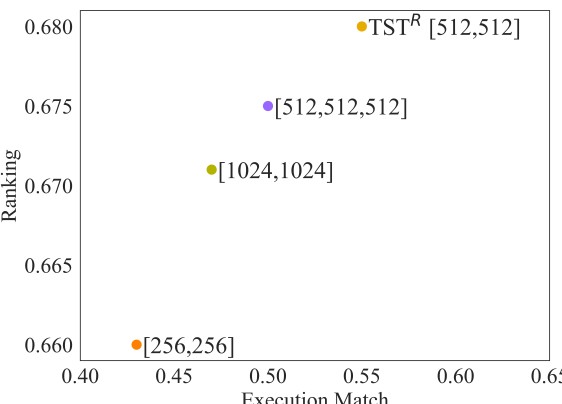

Figure 3: Relation between execution match (top-1) and our pairwise ranking (•) for different fully connected layer configurations of $TST^R$ on the M dataset. There is a clear relation between both metrics.

and $(u_p, u_n)$ coming from train–train, test–train and test–test.

Results on • are shown in Table 2. We see that $TST^R$ is significantly less sensitive to language that it has not seen during training (test–train). TST performs best when the samples are from the same distribution as the training corpus (train–train). This shows that the model overfits on the training distribution, as performance does not carry over to other language distributions.

Table 2: Evaluating influence of variations in language. $TST^f$ uses a FCL on top of frozen sentence embeddings.

|         | test–train | train–train | test–test |
|---------|------------|-------------|-----------|
| $TST^R$ | **0.68**   | 0.72        | **0.58**  |
| TST     | 0.57       | **0.90**    | 0.57      |
| $TST^f$ | 0.49       | 0.66        | 0.51      |

### 5.4 Relevant Examples

We show how selecting examples closer to the relevant decision boundary improves training of $TST^R$. As baselines, we select (1) random training pairs, (2) ten times as many random training pairs, and (3) the best $k$ positive examples (highest code similarity) and negative examples at random.

Table 3 highlights the benefit of selecting the right training set configuration. Even with many more training pairs, random sampling performs poorly. Selecting positive samples based on code similarity improves performance—the system sees more desired examples during training. Selecting relevant negative examples, which are close to the

relevant decision boundary, shows the model what to forget and improves training of TST.

Table 3: Influence of sampling $u_p$ and $u_n$ for training.

| sampling         | M    | SMCF | Bash |
|------------------|------|------|------|
| random           | 0.46 | 0.23 | 0.15 |
| random $\times$ 10 | 0.49 | 0.22 | 0.17 |
| positive only    | 0.61 | 0.35 | 0.20 |
| $TST^R$          | 0.68 | 0.67 | 0.42 |

## 6  Conclusion

We introduce $TST^R$ as a practical improvement of TST for selecting relevant examples in code generation from natural language. $TST^R$ replaces a fine-tuned SentenceBERT model with a small, trainable transformation on top of a frozen embedding model, and provides a strategy for selecting better training examples. Additionally, we show that TST can be evaluated on pairs of utterances from the training set that are ranked with respect to a reference utterance from the testing set, which does not require end-to-end code generation.

Our experiments show that $TST^R$ outperforms classical TST when the language distribution of the example bank does not match that of the tests, that selecting examples closer to the relevant decision boundary improves performance, and that a pairwise ranking evaluation correlates to end-to-end code generation performance.

## 7  Limitations

TST assumes that similar code makes for good examples, and this assumption directly transfers to $TST^R$. When the code is similar overall, but specific details are omitted, this can still result in suboptimal examples.

An additional call to an embedding model or endpoint is required to select relevant examples. Whereas embedding calls are generally cheap[3], the network overhead can cause lower latency than inference with a small transformer.

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
