# OpenReview forum: "TSTR: Target Similarity Tuning Meets the Real World"
_EMNLP/2023/Conference — EMNLP 2023 Findings_

### Official Review · Reviewer_pWu8 · 2023-07-20

**Soundness:** 3

**Excitement:**

4: Strong: This paper deepens the understanding of some phenomenon or lowers the barriers to an existing research direction.

**Paper Topic And Main Contributions:**

This paper studies a practical real-world problem of deploying TST model in production.
It aims to solve 4 practical problems sensitivity and inference cost of transformer models, data set curation for training and final evaluation without code generation.
The paper proposes to train a neural network on top of embeddings to capture code similarity with the TST objective. It then presents a way to select pos/neg examples to yield better results and smaller data sizes for faster training. Finally, they claim the evaluation based on the ranking of examples matches the end-to-end code generation performance.
Overall, I think the contribution is useful but the presentation of this paper needs a lot of work..

**Questions For The Authors:**

What is the R stands for in TSTR?

**Reasons To Accept:**

(1) practical real-world problem with simple techniques to improve performance and present an indirect evaluation protocol for fast development

**Reasons To Reject:**

(1) Training and Evaluation data is not public but I think the main idea is clear and can be extended to other public datasets. Also, the dataset is referenced in another paper so its not this paper's responsibility to release the data.
(2) The presentation is not clear and really needs a lot of work.

**Reproducibility:**

2: Would be hard pressed to reproduce the results. The contribution depends on data that are simply not available outside the author's institution or consortium; not enough details are provided.

**Reviewer Confidence:**

2: Willing to defend my evaluation, but it is fairly likely that I missed some details, didn't understand some central points, or can't be sure about the novelty of the work.

**Typos Grammar Style And Presentation Improvements:**

L020: we how to
L87: our method proposed method
L137: on "Synchromesh" to further
I think the writing needs to be improved.

---

> ### Author Rebuttal · Authors · 2023-08-29
>
> We thank the reviewer for their feedback. We address the reasons for rejection and questions by the reviewer below:
>
> ### Question-1: Training and Evaluation data is not public
>
> Our work uses the data from a prior work [1]. We hope that the authors will release their data in the future.
> To further address this issue, we will add the results on two public datasets: [nl2bash](https://github.com/TellinaTool/nl2bash/tree/master/data) and [SMCalFlow](https://microsoft.github.io/task_oriented_dialogue_as_dataflow_synthesis). Preliminary results on nl2bash show that their command match score improves by 50% between random few-shot and theoretical TST maximum. Additionally, see below for a private release of the data and code.
>
> ###  Question-2: The presentation is not clear and really needs a lot of work
>
> Thank you for your feedback. We will (1) clarify details on training set collection, (2)      clarify the contributions, (3) improve justification of the proxy metric, (4) improve the flow of the technical section, and (5) add results from two new domains.
>
> ###  Question-3: What is the R stands for in TSTR?
>
> The R stands for **r**eal-world, relating to the practical improvements we make to train, evaluate and deploy TST in real scenarios.
>
> ### Data and Code
>
> For review purposes, we privately share data and code [here](https://figshare.com/s/a8c3dc246a3d7d95a9e5). It is not meant to be shared publicly. All contained resources are proprietary.
>
>
> ### References
>
> [1] Anirudh Khatry, Joyce Cahoon, Jordan Henkel, Shaleen Deep, Venkatesh Emani, Avrilia Floratou, Sumit Gulwani et al. “From Words to Code: Harnessing Data for Program Synthesis from Natural Language.” arXiv preprint arXiv:2305.01598 (2023).

---

### Official Review · Reviewer_TieY · 2023-08-02

**Soundness:** 4

**Excitement:**

3: Ambivalent: It has merits (e.g., it reports state-of-the-art results, the idea is nice), but there are key weaknesses (e.g., it describes incremental work), and it can significantly benefit from another round of revision. However, I won't object to accepting it if my co-reviewers champion it.

**Paper Topic And Main Contributions:**

In this paper the authors propose several modifications to Target Similarity Tuning (TST).
TST consists of fine-tuning a sentence embedding model (SBert) such that the similarity between utterance (commands) embeddings are close to the similarity between the associated code examples.

The authors propose fine-tuning only a fully-connected network that receives as input the sentence embeddings instead of fine-tuning the embedding model. This allows the use of larger and third-party LLMs.

The authors also propose to filter the dataset such that it contains more relevant examples (examples for which the utterance similarity is high).

Lastly, the authors proposed a simpler metric that does not require the code snippets to be run. Instead, they construct a dataset using a utterance from the test set and two target utterances sampled from the training set. And see the proportion of correct decisions.


**Reasons To Accept:**

- The proposed modifications lead to gains, especially when the language distribution of the examples is different than the test set one.
- The dataset filtering improves results while making the train faster.
- The proposed evaluation method is simpler and correlates well with the end to end evaluation.


**Reasons To Reject:**

- This work is very incremental.


**Reproducibility:**

4: Could mostly reproduce the results, but there may be some variation because of sample variance or minor variations in their interpretation of the protocol or method.

**Reviewer Confidence:**

3: Pretty sure, but there's a chance I missed something. Although I have a good feel for this area in general, I did not carefully check the paper's details, e.g., the math, experimental design, or novelty.

---

> ### Author Rebuttal · Authors · 2023-08-29
>
> We thank the reviewer for their feedback. We address the reasons for rejection and questions by the reviewer below:
>
> ### Question-1: This work is very incremental
>
> This work builds on the concept of TST and **improves** on important implementational details for **training** (selecting examples and allowing synthetic data generation), **evaluating** (cheap evaluation with a proxy metric) and **deploying** (API call and small FCNN do not have hardware requirements) TST in practice.
>
> We will report results on two additional (public) datasets to further demonstrate these contributions ([nl2bash](https://github.com/TellinaTool/nl2bash/tree/master/data) and [SMCalFlow](https://microsoft.github.io/task_oriented_dialogue_as_dataflow_synthesis)).
>
> ### Data and Code
>
> For review purposes, we privately share data and code [here](https://figshare.com/s/a8c3dc246a3d7d95a9e5). It is not meant to be shared publicly. All contained resources are proprietary.

---

### Official Review · Reviewer_cbui · 2023-08-05

**Soundness:** 3

**Excitement:**

3: Ambivalent: It has merits (e.g., it reports state-of-the-art results, the idea is nice), but there are key weaknesses (e.g., it describes incremental work), and it can significantly benefit from another round of revision. However, I won't object to accepting it if my co-reviewers champion it.

**Paper Topic And Main Contributions:**

In this paper, the authors try to improve the Target Similarity Tuning (TST) application on the Code Generation task.
They improve the similarity of the sentence embedding to match the code similarity and train the TST model with a small train set.
They also introduce a ranking-based evaluation for TST, without the requirement of expensive end-to-end code generation experiments.

**Reasons To Accept:**

This proposed method could enhance the application of TST in the real world, by reducing the sensitivity of language and enabling the inference with transformer models.
Besides, the proposed method could reduce the requirement for the scale of the training set.
Further, the authors also propose an evaluation method for the TST task.

**Reasons To Reject:**

1. Lack of comparison with other similar methods.
2. The metric of evaluation needs a clearer explanation for realistic meaning.
3.  Lack of description for the training set, which is one of the issues to be solved in this paper.

**Reproducibility:**

3: Could reproduce the results with some difficulty. The settings of parameters are underspecified or subjectively determined; the training/evaluation data are not widely available.

**Reviewer Confidence:**

4: Quite sure. I tried to check the important points carefully. It's unlikely, though conceivable, that I missed something that should affect my ratings.

---

> ### Author Rebuttal · Authors · 2023-08-29
>
> We thank the reviewer for their feedback. We address the reasons for rejection and questions by the reviewer below:
>
> ### Question-1: Lack of comparison with other similar methods.
>
> This work focuses specifically on the in-context (few-shot) learning problem. In this setting, common methods in practice are to select random examples [1], use embeddings directly [4] or fine-tune a separate transformer model [2,3] to select the few-shot examples. We compare to these approaches. Recent work rewrites the input to the embedding model to obtain embeddings that more closely align to a given task [5]. We will compare to this method as well.
> Furthermore, we will improve our evaluation by adding results on public NL to Code datasets: [nl2bash](https://github.com/TellinaTool/nl2bash/tree/master/data) and [SMCalFlow](https://microsoft.github.io/task_oriented_dialogue_as_dataflow_synthesis). SMCalFlow was used to evaluate TST [3].
>
> ### Question-2: The metric of evaluation needs a clearer explanation for realistic meaning.
> We use the execution match metric to report results for the most important question: end-to-end code-generation accuracy (Figure 2). This metric is popular in text-to-code settings [1,2,3,6,7] as a problem can be solved by multiple programs.  Because end-to-end testing with few-shot prompts is expensive, we use our proposed pairwise  ranking as a cheaper proxy metric for other questions (language variation and boundary example selection). Our experiments show that this proxy metric is correlated well with end-to-end results.  We can report end-to-end results for these questions on all datasets (see Question-1) in our revision.
>
> ### Question-3: Lack of description for the training set, which is one of the issues to be solved in this paper.
>
> Thanks for your feedback. We collected the training set by scraping M code from the community channel set up by Power Query [8]. The natural language utterances for the M code were generated by using an LLM. This process used a few-shot prompt using the power query documentation as reference [9]. We will extend our discussion in Section 4 (under Data and Model) to include more details.
>
> ### Data and Code
>
> For review purposes, we privately share data and code [here](https://figshare.com/s/a8c3dc246a3d7d95a9e5). It is not meant to be shared publicly. All contained resources are proprietary.
>
> ### References
>
> [1] Ansong Ni, Srini Iyer, Dragomir Radev, Ves Stoyanov, Wen-tau Yih, Sida I. Wang, Xi Victoria Lin. “LEVER: Learning to Verify Language-to-Code Generation with Execution.” ICML 2023.
>
> [2] Naman Jain, Skanda Vaidyanath, Arun Iyer, Nagarajan Natarajan, Suresh Parthasarathy, Sriram Rajamani, and Rahul Sharma. “Jigsaw: Large language models meet program synthesis.” ICSE 2022.
>
> [3] Gabriel Poesia, Alex Polozov, Vu Le, Ashish Tiwari, Gustavo Soares, Christopher Meek, and Sumit Gulwani. “Synchromesh: Reliable Code Generation from Pre-trained Language Models.” ICLR 2021.
>
> [4] Jiachang Liu, Dinghan Shen, Yizhe Zhang, Bill Dolan, Lawrence Carin, and Weizhu Chen. “What Makes Good In-Context Examples for GPT-3?” DeeLIO 2022 (ACL Workshop).
>
> [5] Shengnan An, Bo Zhou, Zeqi Lin, Qiang Fu, Bei Chen, Nanning Zheng, Weizhu Chen, and Jian-Guang Lou. “Skill-Based Few-Shot Selection for In-Context Learning.” arXiv preprint arXiv:2305.14210 (2023).
>
> [6] Qinkai Zheng, Xiao Xia, Xu Zou, Yuxiao Dong, Shan Wang, Yufei Xue, Lei Shen et al. “CodeGeeX: A Pre-Trained Model for Code Generation with Multilingual Benchmarking on HumanEval-X.” KDD 2023.
>
> [7] Yujia Li, David Choi, Junyoung Chung, Nate Kushman, Julian Schrittwieser, Rémi Leblond, Tom Eccles et al. “Competition-level code generation with alphacode.” Science 2022.
>
> [8] Power Query Community Forum. https://community.fabric.microsoft.com/t5/Power-Query/bd-p/power-bi-services
>
> [9] Power Query Documentation. https://learn.microsoft.com/en-us/powerquery-m/

---

### Meta-Review · Area_Chair_Q6B8 · 2023-09-19

**Recommendation:** 2

**Metareview:**

The main concern raised by the reviewer are that the code and data are not publicly available. The authors did make the code and data privately available for review but the reviewers felt that is not enough (and I agree). Apart from this the paper needs a few changes:

1. Need to report results on public datasets [the authors do promise that they will report results on 2 public datasets but the results on only 1 were provided during the response period)
2. More details on the manner in which the training dataset was curated.
3. Comparison with more recent related work (such as [5] as suggested by the authors)
4. The presentation could be better.

I request the authors to address the above concerns in a subsequent revision.

---

### Decision · Program_Chairs · 2023-10-07

**Decision:**

Accept-Findings

**Comment:**

The main concern raised by the reviewer are that the code and data are not publicly available. The authors did make the code and data privately available for review but the reviewers felt that is not enough (and I agree). Apart from this the paper needs a few changes:

1. Need to report results on public datasets [the authors do promise that they will report results on 2 public datasets but the results on only 1 were provided during the response period)
2. More details on the manner in which the training dataset was curated.
3. Comparison with more recent related work (such as [5] as suggested by the authors)
4. The presentation could be better.

I request the authors to address the above concerns in a subsequent revision.